# SiO_2_ Microsphere Array Coated by Ag Nanoparticles as Raman Enhancement Sensor with High Sensitivity and High Stability

**DOI:** 10.3390/s22124595

**Published:** 2022-06-17

**Authors:** Haiyang Sha, Zhengkun Wang, Jie Zhang

**Affiliations:** The Key Laboratory of Optoelectronic Technology & System (Ministry of Education), Chongqing University, Chongqing 400044, China; 202008021055t@cqu.edu.cn (H.S.); 20190801173@cqu.edu.cn (Z.W.)

**Keywords:** Raman scattering, SiO_2_ microsphere array, Ag nanoparticles

## Abstract

In this paper, a monolayer SiO_2_ microsphere (MS) array was self-assembled on a silicon substrate, and monolayer dense silver nanoparticles (AgNPs) with different particle sizes were transferred onto the single-layer SiO_2_ MS array using a liquid–liquid interface method. A double monolayer “Ag@SiO_2_” with high sensitivity and high uniformity was prepared as a surface-enhanced Raman scattering (SERS) substrate. The electromagnetic distribution on the Ag@SiO_2_ substrate was analyzed using the Lumerical FDTD (finite difference time domain) Solutions software and the corresponding theoretical enhancement factors were calculated. The experimental results show that a Ag@SiO_2_ sample with a AgNPs diameter of 30 nm has the maximal electric field value at the AgNPs gap. The limit of detection (LOD) is 10^−16^ mol/L for Rhodamine 6G (R6G) analytes and the analytical enhancement factor (AEF) can reach ~2.3 × 10^13^. Our sample also shows high uniformity, with the calculated relative standard deviation (RSD) of ~5.78%.

## 1. Introduction

Surface-enhanced Raman scattering (SERS) is powerful analytical technology with the advantages of high sensitivity, trace detection, and non-destructive detection. It has been widely used in the fields of chemistry, physics, optics, and material science [1,2,3].

For the SERS enhancement mechanism, two mechanisms have been proposed: electromagnetic enhancement (EM) and chemical enhancement (CM) [4,5]. The former is due to the localized surface plasmon resonance (LSPR) of metal structures [6]. The latter is attributed to the surface chemisorption, surface complexes, and charge transfer between the analyte and the metal surface [7,8]. EM plays a dominant role in Raman enhancement [9]. 

Common SERS substrates are mainly precious metals. Among them, the precious metals gold and silver have been extensively studied. Bimetallic nanostructures have been extensively studied in recent years. For example, J. W. Liu et al., reported a simple microbial synthesis method for the manufacture of Au@Ag nanoislands for a quantitative SERS detection [10]. G. J. Weng et al., showed that the paper substrate doped with Au nanoparticles prepared using an inkjet printing method could significantly improve SERS performance, and it was reproduced by the secondary growth of Ag nanoparticles [11]. However, due to the existence of van der Waals forces and high surface energy between metal nanoparticles, they can easily agglomerate during synthesis and use [12]. For example, silver nanoparticles with particle size less than 20 nm are easy to agglomerate in practical use due to their high surface energy, which reduces their various application performances [13]. In order to solve the problem of activity and stability, polymers, carbon and oxides as substrates have been widely used to solve the problem of agglomeration of small size nanoparticles [14]. Hybrid systems consisting of polymer-nanoparticle or inorganic material-nanoparticle arrangements have been developed. Among them, metal nanoparticles can provide the generated nanoparticles with unique properties [15,16,17] and the polymer or inorganic materials, as the carriers can control the spatial distribution of nanoparticles [18,19,20].

As a template with simple preparation and a relatively low price, microspheres can obtain large-area uniform periodic structure through a self-assembly technology, which has been widely used in SERS research. J. F. Chen et al., prepared three-dimensional self-assembled polystyrene (PS) microspheres/ultra-thin silver film composite structure with excellent SERS performances (enhancement factor (EF ≈ 10^8^)) and uniformity (relative standard deviation (RSD ≈ 8%)) [21]. However, for PS microspheres, the low melting point affects performance in some high-temperature applications. A core–shell system immobilized Ag NP via chemical reduction of Ag^+^ with sodium borohydride, which was proposed for catalytic applications that can be adjusted with temperature changes [22]. However, in this case, non-biocompatible polymers were used, hindering biomedical applications. Therefore, in order to overcome this barrier, metal-coated SiO_2_ particles prepared with different synthesis methods have been reported. H. M. Kim et al., produced a SERS substrate of silica microspheres embedded by silver nanoparticles, and the enhancement factor reached 10^10^ [23]. H. J. Chang et al., prepared a Ag@SiO_2_ substrate using an in situ manufacturing method and completed the limit of detection for prostate-specific antigen (PSA) to 0.11 pg/mL using a regional scanning method [24]. X. H. Pham et al., assembled an alloyed NP immobilized SERS active substrate (SiO_2_@Au@Ag NPs). This nanoprobe showed significantly enhanced SERS characteristics, in which the “hot spot” could be controlled by changing the concentration of Ag^+^, realizing the detection of 2.4 nM 4–aminothiophenol (ATP) [25]. Although these materials offer significant advantages, it is still possible to develop an easier and lower-cost method to prepare SERS substrates with higher Raman-enhancement performance.

In this paper, we prepared monolayer AgNPs with high uniformity using a liquid–liquid interface method. These were then transferred to self-assembled monolayer SiO_2_ microspheres, forming a double monolayer Ag@SiO_2_ SERS substrate. The characteristics of Raman enhancement are discussed in detail.

## 2. Materials and Methods

### 2.1. Materials and Characterization Equipment

The silicon wafer was P-type (100) single-throwing hydrogen peroxide type (1~10 Ω.cm) (Zhejiang Lijing Optoelectronics Technology Co., Ltd., Hangzhou, China), surface hydroxylated SiO_2_ MS with a diameter of 600 nm (Zhongke Leiming Bio Medical Nanotechnology Company, Beijing, China), Rhodamine 6G (Shanghai Aladdin Co., Ltd., Shanghai, China), and crystal violet (CV, Shanghai Aladdin Co., Ltd., Shanghai, China).

The sample’s geometrical characterization was measured using a Quattro S field emission environmental scanning electron microscope (SEM). UV–visible spectra were acquired with a Shimadzu UV–3600 spectrophotometer (Kyoto, Japan). Raman spectra were collected with a laser confocal Raman spectrometer (Horiba JY LabRAM HR Evolution, France), equipped with a 100× objective lens, a numerical aperture (NA) of 0.9, and a work distance (WD) of 0.21 mm. An air-cooled double-frequency Nd:Yag green laser was used as the incident light, with the wavelength of 532 nm and the power of 5 mW to avoid sample heating and photo–induced damage. An integration time of 2 s was used in all measurements. The laser spot size was 1.2 μm. Before Raman testing, 5 μL of the solution to be tested was dripped onto the substrate and then dried naturally for measurement. A baseline correction was performed for the SERS measurements to eliminate a large background that correlates with the Raman intensity. In order to reduce random error, each Raman signal was averaged from multiple measurements.

### 2.2. Preparation of the SERS Substrate

The preparation process of the SERS substrate comprises three steps and is shown in Figure 1. The first step is the preparation of silver sol with different particle sizes; the second step is the preparation of the monolayer SiO_2_ MS; and the third step is the preparation of the monolayer Ag@SiO_2_ substrate.

#### 2.2.1. Preparation of Silver Sol with Different Particle Sizes

A total of 170 mg polyvinylpyrrolidone (PVP, k30) and 170 mg AgNO_3_ was added, in that order, to 40 mL deionized water and stirred to dissolve. Then, 0.4 mL 5 mol/dm^3^ NaCl was added and stirred magnetically for 15 min in the dark to obtain AgCl colloid.

A 2.8 mL 0.5 mol/dm^3^ NaOH solution was added into 20 mL 0.05 mol/dm^3^ Ascorbic acid (AA) solution and mixed evenly. Then, 2.5 mL AgCl colloid prepared in advance was added and stirred for 2 h in a dark environment. Thirty nm silver sol was obtained via multiple centrifugation and dispersion. Fifty nm silver sol could also be made in the same way by changing the amount of NaOH to 2.5 mL.

The silver sol of 100 nm, 150 nm, and 200 nm was grown from the seed of 50 nm silver sol. The steps were as follows: (a) We added 100 mg PVP into a 20 mL 0.05 mol/dm^3^ AA solution, then added 3570, 960, and 400 μL 50 nm silver sol, respectively, and kept stirring; (b) 0.5 mL 0.5 mol/dm^3^ NaOH and 2.5 mL AgCl colloid were added, in that order, and stirred continuously under dark conditions for 16 h; (c) the Ag sol was obtained via additional multiple centrifugation and dispersion in order to remove residual organic material.

#### 2.2.2. Preparation of Monolayer SiO_2_ MS

A silicon wafer was placed in acetone and anhydrous ethanol for ultrasonic cleaning for 15 min to remove organic solvent impurities on the surface. The silicon wafer was then placed in concentrated H_2_SO_4_/H_2_O_2_ (volume ration of 2.4:1) and heated in a constant-temperature water bath at 90 °C for 1 h. Finally, the silicon wafer was rinsed with deionized water 3 to 4 times to remove the acid and then dried with a nitrogen gun. 

A pipette gun was used to transfer the suspension of silica microspheres to hydrophilic silicon wafers, and a multi-stage rotary coater mode was selected. The first stage was glue dropping, with a speed of 500 r/min and a duration of 15 s. The second stage was homogenization, with a speed of 1800 r/min and a duration of 60 s. After placing and drying, a self-assembled monolayer SiO_2_ MS was obtained (Figure 1b).

#### 2.2.3. Preparation of Monolayer Ag@SiO_2_ Substrate

Firstly, we added 5 mL hexane into a beaker containing 5 mL silver sol to form an interface. Then, 0.5 mL of 0.1 mmol/L Mercaptopropyl trimethoxysilane (MPTMS) was added to the hexane layer. Thirdly, a syringe was used to slowly add ethanol droplets to the interface, and AgNPs were transferred to the interface (Figure 1a). After the volatilization of hexane, the monolayer SiO_2_ MS template was inserted at a 45° obliquity under the liquid surface, then raised vertically and dried to obtain a double monolayer Ag@SiO_2_ substrate (Figure 1b).

## 3. Results

### 3.1. SEM Characterization

The SEM images in Figure 2(a1–e1) show the dense distribution of monolayer AgNPs using the liquid–liquid interface method. The calculated particle sizes in Figure 2(a2–e2) are 32 nm, 53 nm, 104 nm, 150 nm, and 197 nm, respectively. SEM images of samples Ag@SiO_2_ are shown in Figure 2(a3–e3). It can be seen that SiO_2_ MS can be wrapped well by smaller AgNPs with a size of 30 nm and 50 nm, and the corresponding Ag@SiO_2_ samples have better uniformity compared with those coated by larger AgNPs of 104 nm, 150 nm, and 197 nm. An increase in Ag particle size, especially when the particle size increases to 104 nm, 150 nm, and 197 nm, makes it difficult for AgNPs to form a dense monolayer on the microspheres, probably due to the large mass and size. In order to explore the driving force that caused the SiO_2_ MS to be “wrapped” by AgNPs, we tested the electrification of the prepared silver sol using the Zeta potential analyzer and found the Zeta potential of silver sol was around −8.5 mV. It was negatively charged, while the SiO_2_ MS was electrically neutral. Therefore, we concluded that under the action of electrostatic adsorption, when the particle size of SiO_2_ MS is much larger than that of AgNPs, the former can be well wrapped by the latter. When the particle size of SiO_2_ MS is a little larger than that of AgNPs, it cannot be well wrapped, resulting in poor uniformity of Ag distributions (Figure 2(c3–e3)). From the SEM images (Figure 2(a3–e3)), the brightness of AgNPs and SiO_2_ MS is different. So, we binarized the SEM images to calculate the coverage of AgNPs. The calculated Ag coverage rate of Ag@SiO_2_ was 68.6%, 77.6%, 46.8%, 40.9%, and 48.5%, respectively. Additionally, due to the single-layer structure and the uniformity of Ag distribution, the Ag coverage rate on one microsphere was almost the same as that on microspheres. Finally, based on the enhancement mechanism of SERS substrates, samples Ag@SiO_2_ with Ag size of 30 nm and 50 nm (Ag@SiO_2_-30, Ag@SiO_2_-50) were chosen for Raman measurements. The calculated gap between AgNPs of these samples was roughly between 0.5~2 nm (Figure 2(f1–g2)), based on “Nano Measurer” software.

### 3.2. UV–Vis Absorption

Figure 3a shows the absorption spectra of Ag@SiO_2_ samples with different AgNPs sizes. With the increase in Ag size, the absorption peak shows a red shift from ~475 to ~562 nm and broaden phenomenon, which is basically consistent with the change of absorption spectrum of pure AgNPs [26]. For the curve of 30 nm, we can see that it is a little different from the curve of other particle sizes. It is generally believed that because of its small size, the radiation loss is small, which leads to a sharper resonance peak [27]. We also concluded that the deeper dip is due to more AgNPs on a single SiO_2_ MS, which leads to a stronger interaction between the cavity mode of SiO_2_ MS and the radiation mode of AgNPs, and the more pronounced spatial spectrum overlap [28]. It is generally believed that the reason for the SERS effect is the generated LSPR induced by the absorption of incident light by metal nano-structured materials. Therefore, a 532 nm laser was selected as the excitation light source during our Raman measurements so as to achieve a high Raman-enhancement performance.

### 3.3. Raman Measurements

#### 3.3.1. FDTD Simulation

Three-dimensional FDTD (finite difference time domain) Solutions are used to study the spatial distribution of electronic field (E-field). Based on the SEM images in Figure 2(g1,g2), in our models, AgNPs with diameters of 30 nm and 50 nm, and gaps of 0.5 nm, 1 nm, and 2 nm were selected for simulation, respectively. In our experiment, the surface of SiO_2_ MS was hydroxylated, and the refractive index was higher than that of ordinary SiO_2_, setting *n* = 1.65. The optical properties of Ag were taken from Palik [29] and the refractive index was set to the default Palik (0~2 µm) in the FDTD material library. The incident laser was a plane wave with a wavelength of 532 nm, and the incident electric field (E_0_ = 1 V/m) with Y-polarization propagated along the −Z direction. The surrounding medium was set as air, and the perfect matching layer (PML) was used as the boundary condition, as shown in Figure 4a–d.

For samples with AgNPs size of 30 nm, shown in Figure 4(c1–c3) (X-normal monitor), Figure 4(c4–c6) (Z-normal monitor), a strong E-field was located at the gap between AgNPs, called the “hot spot”. The maximal E-field intensity was 700 V/m, 190 V/m, and 37 V/m at gap of 0.5 nm, 1 nm, and 2 nm, respectively. For samples with AgNPs size of 50 nm, the corresponding maximal E-field was 500 V/m, 250 V/m, and 160 V/m, respectively (Figure 4(d1–d3) (X-normal monitor), Figure 4(d4–d6) (Z-normal monitor)).

Based on the simulation results, the theoretical enhancement factor can be calculated by
(1)$$EF=\frac{{\left|E_{out}(\omega_{0})\right|}^{2}{\left|E_{out}(\omega_{s})\right|}^{2}}{{\left|E_{0}\right|}^{4}}\approx \frac{{\left|E_{out}\right|}^{4}}{{\left|E_{0}\right|}^{4}}$$
where, $E_{0}$ = 1 V/m is the intensity of an incident electric field, $E_{out}(\omega_{0})$ and $E_{out}(\omega_{s})$ are the electric field intensity of at frequency of $\omega_{0}$ and Raman scattered light frequency of $\omega_{s}$, respectively. Therefore, the theoretical maximal EF value is ~2.4 × 10^11^ (30 nm AgNPs with a gap of 0.5 nm) and ~6.2 × 10^10^ (50 nm AgNPs with a gap of 0.5 nm), respectively.

As for these two samples, the electric field decreases with the increase in the gap, shown in Figure 4e. In order to gain a higher E-field, a larger AgNPs coverage with a smaller gap on the surface of SiO_2_ MS array should be undertaken.

#### 3.3.2. Raman Measurements with Different R6G Concentrations 

In order to study the Raman enhancement effect of Ag@SiO_2_ substrate, R6G was used as the probe molecule. Figure 5a–d shows the Raman spectra of R6G with different concentrations from 10^−12^ mol/L to 10^−16^ mol/L absorbed on our samples. Obvious Raman peaks of R6G are clearly visible. The limit of detection is down to 10^−16^ mol/L (Figure 5e). Among our samples, samples Ag@SiO_2_–30 and Ag@SiO_2_–50 show better Raman enhancement effects. The analytical enhancement factor (AEF) is always used to evaluate the Raman enhancement effect of SERS substrates, which is calculated as follows.
(2)$$\mathrm{AEF}=\frac{I_{SERS}/c_{SERS}}{I_{RS}/c_{RS}}$$
where, $I_{SERS}$ and $I_{RS}$ is Raman intensity under SERS and non-SERS condition, respectively. $c_{SERS}$ and $c_{RS}$ represent the analyte concentration under SERS and non-SERS, respectively. In our experiment, $I_{RS}$ is the Raman intensity of R6G solution with concentration of 10^−^^2^ mol/L on clean silicon wafer.

Figure 5f shows the variation trend of Raman intensity of 10^−16^ mol/L R6G with particle sizes at 1650 cm^−1^ and the calculated experimental AEF. AEF values of sample Ag@SiO_2_-30 can reach ~2.3 × 10^13^. Among the samples, sample Ag@SiO_2_–30 shows the best enhancement property. In order to investigate the function of SiO_2_ MS in composite structure, a Raman measurement on monolayer Ag on silicon was carried out under the same measurement condition shown in Figure 5g. With a R6G concentration of 10^−16^ mol/L, the Raman intensity at 610 cm^−1^ on sample Ag@SiO_2_ is 2.4 times than that of Ag on silicon, and the enhancement is more obvious under a higher concentration. Herein, SiO_2_ MS has multiple reflections of the incident light, which could contribute to the Raman enhancement [30]. Besides our study, many methods for preparing Ag@SiO_2_ SERS substrates were reported. C. Liu et al., prepared the ordered silica microsphere template using vertical evaporation technology, and the Ag(NH_3_)_2_^+^ adsorbed on the microspheres by an electrostatic action was reduced to Ag nanoparticles by ethanol, forming a Ag@SiO_2_ SERS substrate. The LOD is 10^−^^8^ mol/L for Nile blue A (NBA) analytes and the AEF can reach ~10^5^ [31]. J. F. Li et al., used poly vinyl pyrrolidone (PVP) as a modifier and reducing agent, deposited silver nanoparticles on the surface of SiO_2_ MS in situ, prepared silver-modified SiO_2_ MS, and formed a Ag@SiO_2_ array via evaporation-induced self-assembly, which realized the detection of R6G with concentrations as low as 10^−^^11^ mol/L [32]. Similarly, J. Huang et al., used 3–Aminopropyl triethoxysilane (APTES) as a modifier and PVP as a reducing agent to deposit silver nanoparticles on the surface of SiO_2_ MS, prepared Ag@SiO_2_, and the LOD to R6G was 10^−^^8^ mol/L [33]. Compared with other SiO_2_ MS composites, our samples with the double monolayer structure have higher Raman enhancement performance with an LOD of 10^−16^ mol/L for R6G analytes, and an AEF of ~2.3 × 10^13^. The average number of molecules in the laser spot can be estimated by carefully calculating the detection volume, liquid spot, and concentration of R6G solution on the substrate. The average number of molecules in the detection region for 1 × 10^−16^ mol/L R6G (in solutions) was about 10^−4^ molecules, theoretically. In actuality, the distribution of the analyte was not totally even. During the measurement processing, we always looked for the place where there was a signal and took multiple measurements to gain the average value. In addition, as for the microsphere substrate, the analyte is easily trapped at the gap of AgNPs or MS, so it is helpful to gain enhanced Raman signals [34].

#### 3.3.3. Uniformity

In order to verify the uniformity of SERS substrates, a Raman mapping measurement with an area of 15 µm × 15 µm at a 5 µm step was performed with R6G of 10^−12^ mol/L as the analytes, shown in Figure 6. The uniformity of samples Ag@SiO_2_–100 and Ag@SiO_2_–150 was poor, and the corresponding relative standard deviations (RSD) at 610 cm^−1^ were 26.24% and 31.36%, respectively, as shown in Figure 6a,b. The RSD values were larger than 30% at 770 cm^−1^ and 1650 cm^−1^ for these two samples. Samples Ag@SiO_2_–30 and Ag@SiO_2_–50 had a better uniformity, and RSD values at 770 cm^−1^ were 6.23% and 5.78%, respectively, shown in Figure 6c,d. 

#### 3.3.4. Multi-Molecular Detection

In order to investigate the ability of multi-molecule detection of our samples (Ag@SiO_2_–30), the mixture of R6G (10^−9^ mol/L) and CV (10^−7^ mol/L) was used as the analytes. Shown in Figure 7, Raman peaks of two probe molecules (R6G: 610 cm^−1^, 770 cm^−1^, 1650 cm^−1^, etc., CV: 722 cm^−1^, 913 cm^−1^, 1614 cm^−1^, etc.) can be detected easily, which indicates that the substrate has a good Raman performance in multi-molecule mixed solution.

## 4. Conclusions

In this paper, AgNPs with different diameters were successfully transferred to a self-assembled SiO_2_ MS array via a liquid–liquid interface method. After preparation and Raman measurements were optimized among our samples, sample Ag@SiO_2_-30 showed almost the best Raman enhancement performance, with an AEF of ~2.3 × 10^13^, a LOD of 10^−16^ mol/L and a good uniformity (RSD = 6.23% at 770 cm^−1^). Due to its high sensitivity and good uniformity, this SERS substrate will have a great application in future SERS sensors.

## Figures and Tables

**Figure 1 sensors-22-04595-f001:**
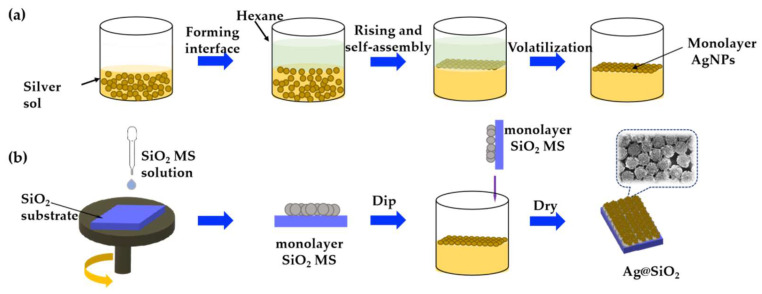
Preparation process of (**a**) monolayer AgNPs and (**b**) Ag@SiO_2_ substrate.

**Figure 2 sensors-22-04595-f002:**
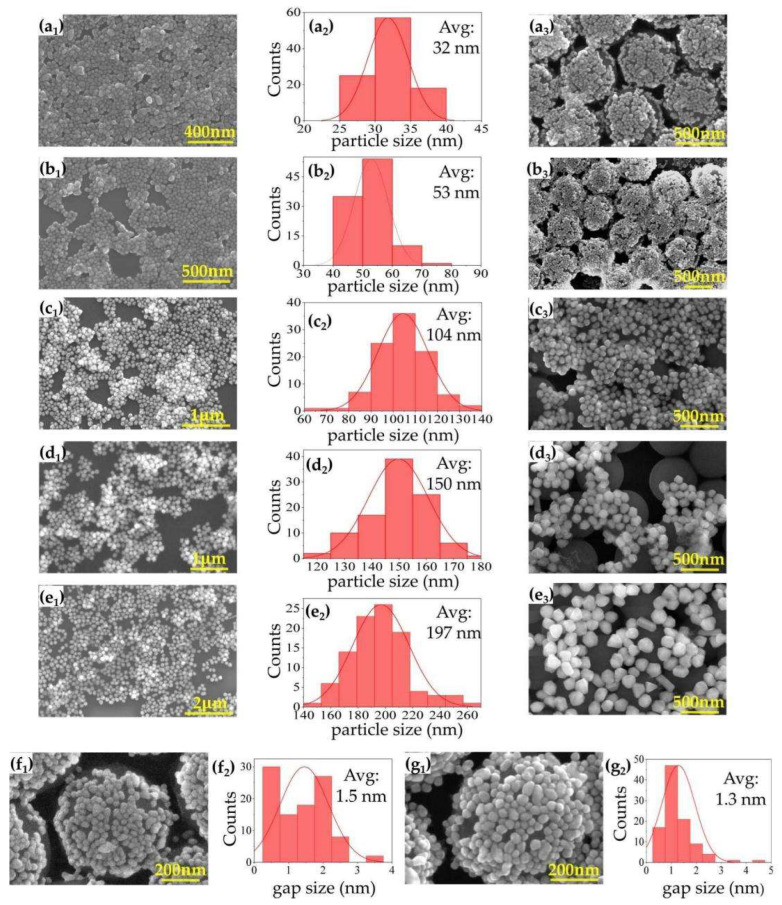
SEM images of prepared Ag with diameters of (**a1**) 30 nm; (**b1**) 50 nm; (**c1**) 100 nm; (**d1**) 150 nm; (**e1**) 200 nm. (**a2**–**e2**) The corresponding calculated Ag particle size distributions for figures (**a1**–**e1**). SEM images of Ag@SiO_2_ with Ag size of (**a3**) 30 nm; (**b3**) 50 nm; (**c3**) 100 nm; (**d3**) 150 nm; (**e3**) 200 nm. (**f1**,**g1**) Enlarged SEM in figures (**a3**,**b3**) and (**f2**,**g2**), and the corresponding calculated gap between AgNPs in (**f1**,**g1**).

**Figure 3 sensors-22-04595-f003:**
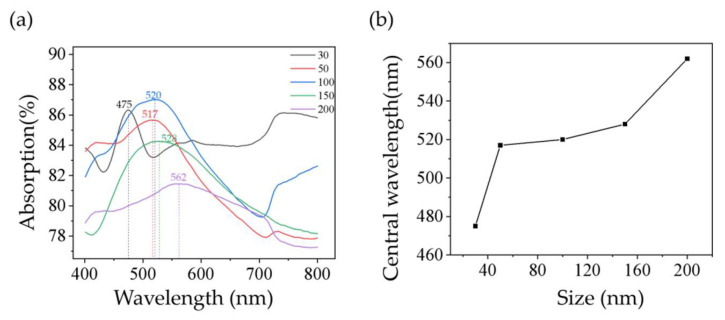
(**a**) Absorption spectra of samples Ag@SiO_2_; (**b**) variation of the absorption peak wavelength with different Ag size (30, 50, 100, 150, and 200 nm).

**Figure 4 sensors-22-04595-f004:**
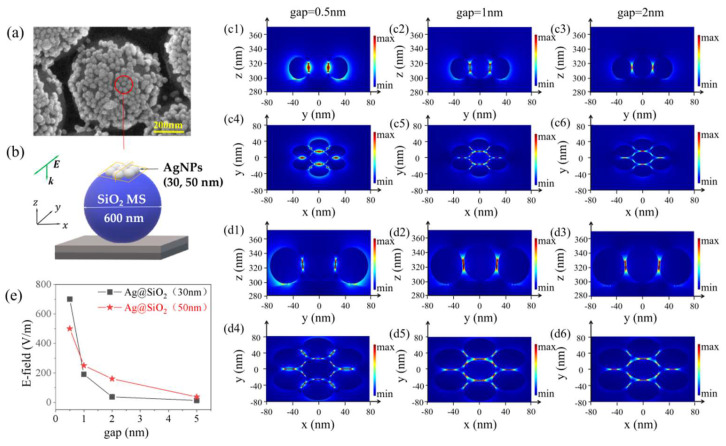
(**a**) SEM of Ag@SiO_2_; (**b**) simulation model: electromagnetic field distribution of Ag@SiO_2_ substrate; (**c**) electromagnetic field distribution of Ag@SiO_2_–30 with different gaps; (**d**) electromagnetic field distribution of Ag@SiO_2_–50 with different gaps; (**e**) the relationship between electric field intensity and gap of AgNPs.

**Figure 5 sensors-22-04595-f005:**
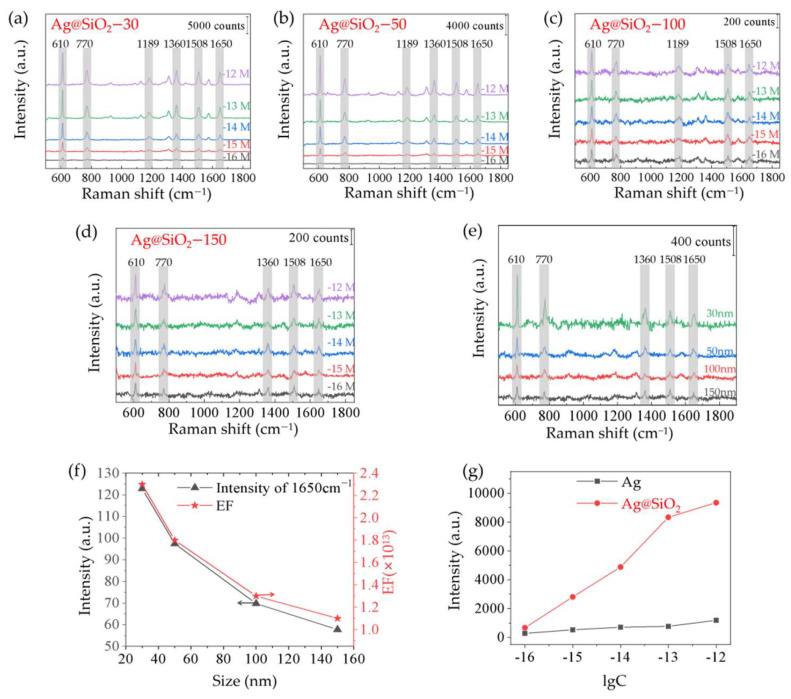
Raman measurements for R6G with different concentration on absorbed on samples with AgNPs diameter of (**a**) 30 nm; (**b**) 50 nm; (**c**) 100 nm; (**d**) 150 nm; (**e**) Raman intensity of 10^−16^ mol/L R6G on different samples; (**f**) Raman intensity 1650 cm^−1^ of 10^−16^ mol/L R6G and the corresponding EF of different samples with different Ag sizes; and (**g**) Raman intensity of 10^−16^ mol/L to 10^−12^ mol/L absorbed on our Ag@SiO_2_ and Ag on silicon.

**Figure 6 sensors-22-04595-f006:**
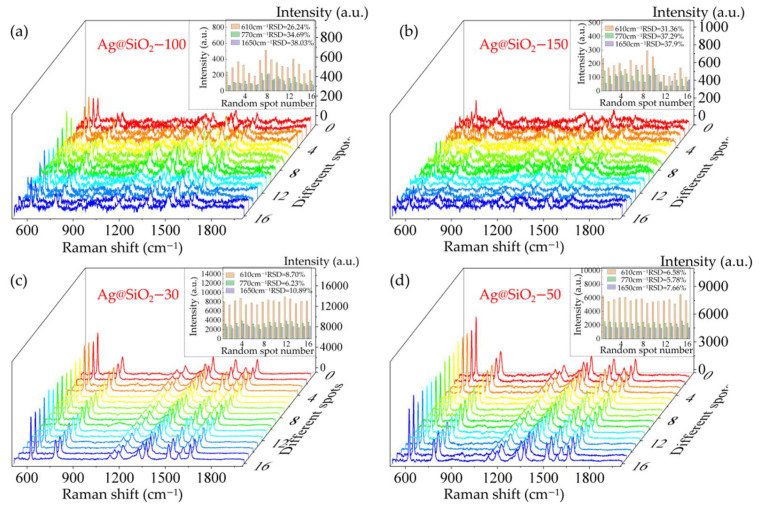
Raman mapping measurements of R6G with concentration of 10^−12^ mol/L absorbed on different samples with Ag diameter of (**a**) 100 nm; (**b**) 150 nm; (**c**) 30 nm; (**d**) 50 nm. The corresponding RSD values at 610 cm^−1^, 770 cm^−1^, and 1650 cm^−1^ were inserted in each figure.

**Figure 7 sensors-22-04595-f007:**
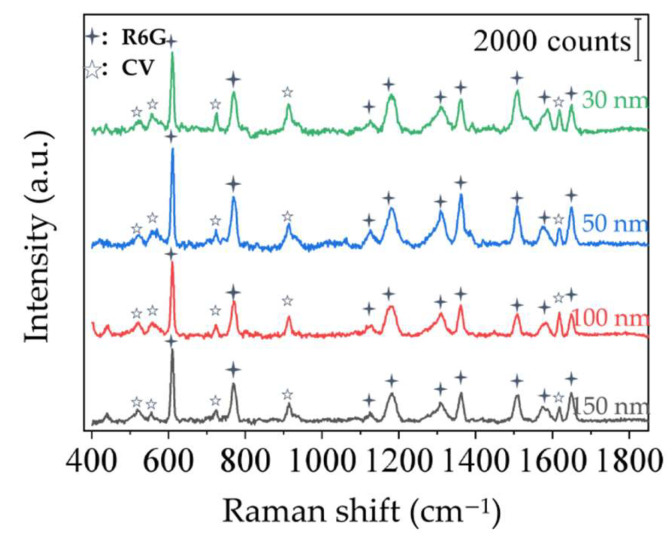
Raman measurement of R6G and CV simultaneously on sample Ag@SiO_2_–30.

## Data Availability

Not applicable.

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
