# Peer review of "SiO2 Microsphere Array Coated by Ag Nanoparticles as Raman Enhancement Sensor with High Sensitivity and High Stability"

_sensors, 2022, doi:10.3390/s22124595_

Round 1
Reviewer 1 Report
The authors report on fabrication of highly sensitive SERS substrates. The proposed technique is based on a monolayer of silica beads covered with silver nanoparticles. Authors use rhodamine 6G to test the sensitivity of the new SERS substrates, which is expressed in terms of the Analytical Enhancement Factor (AEF) and the limit of detection (LOD). Numerical simulations are carried out to support the experimental results.
In general, the paper seems to be interesting. There are, however, some issues that needs to be clarified.
Major problem:
Some of the important experimental details are not presented in the manuscript, which makes the presented results questionable. Authors state that the studied rhodamine 6G samples were dripped onto the substrate and let to dry. The volume of the used samples was not specified. At the same time, the LOD is claimed to be 10-16 mol/L, while the used laser spot size in the Raman instrument was 1.2 µm. Let us assume a realistic scenario: 1 µl of the rhodamine 6G sample was deposited onto the substrate, which (after drying) formed a spot of approx. 4 mm2. In this case, we would have altogether 6.1023x1.10-6x1.10-16=60 molecules of rhodamine 6G on an area of approx. 4 mm2. The probability of finding a single molecule in the laser spot (1.2 µm) is in principle zero (less than 0.00005%). How is it possible, that the signal was detected? Please explain in detail.
Minor issues:
11. Introduction: please add references to the last sentence of the second paragraph: ”Among them, metal nanoparticles…”
22. Please consider using Ag@SiO2 instead of SiO2@Ag.
33. I have concerns regarding Figures f2 and g2. First, the x axis should be “gap size” (not particle size). Second, I think it is not possible to evaluate the gap size distribution from the presented SEM images. Most of the particles seem to touch each other, which would mean that the presented distributions are not correct. I don’t think this experimental gap evaluation is important for the paper. I suggest simply to remove these distributions from the manuscript.
44. Figure 3: panel a) indicates 20nm particles; 30nm is shown in panel b). Which one is correct?
55. What is the reason for the unusual absorption spectrum indicated for 20 nm particles in Figure 3a. Please comment on that.
66. All Figures: increase the font size of the Labels inside the Figures (numbers, text, scale bars, insets, etc.)
77. Figure 4. Enhance the contrast and the quality of the field distribution results. The results are not visible in the present form.
88. Equation (2): How was the IRS signal measured? Give more details.
99. Give the full name of the CV molecule in the text.
Reviewer 2 Report
In this manuscript the authors fabricated Raman enhancement sensors by assembling together monolyaers of Si2O spheres (300 nm the average diameter) and Au nanospheres (with a size varying betwween 30 and 200 nm, respectively). Electron microscopy worp presented is aimed at supporting the assumption that Ag nanoparticles are in contact with SiO2 nanoparticles, but from the micrographs is really hard to find this out. What would be the driving force to lead the SiO2 being 'wrapped" by Ag nanioparticles? How the coverage rate was calculated? Is this coverage rate consistent from a synthesis to another? EDX measurements may help elucidate the ratio between the SiO2 nanoparticles and Ag nanophseres. The use of these sysyems for the detection of R6G and mixtures of R6G and CV is interesting and explained in detail. Overall, the article is interesting and well written; it could be considered for publication into Sensors after the authors address the comments above.
Round 2
Reviewer 1 Report
The authors answered my questions in details. In general, I am satisfied with all the answers and I only have a small comment left:
In their answer to Question 1, point (4) the authors stated : "then (we) look for the place where there is a signal," This is correct, and I still believe this point is critical for measuring the signal at 10e-16M. I suggest to include this sentence (taken from the authors reply) also in the revised manuscript.
